# Composition Dependence Structural and Optical Properties of Silicon Germanium ($Si_\chi Ge_{1-\chi}$) Thin Films

**Syafiqa Nasir** [1], **Fuei Pien Chee** [2,*], **Bablu Kumar Ghosh** [1,*], **Muhammad Izzuddin Rumaling** [2], **Rosfayanti Rasmidi** [3], **Mivolil Duinong** [2] and **Floressy Juhim** [2]

[1] Faculty of Engineering, Universiti Malaysia Sabah, Kota Kinabalu 88400, Sabah, Malaysia; nur_syafiqa_mk20@iluv.ums.edu.my

[2] Faculty of Science and Natural Resources, Kota Kinabalu 88400, Sabah, Malaysia; izzudin123@gmail.com (M.I.R.); mivolilds@yahoo.com (M.D.); floressyj@gmail.com (F.J.)

[3] Faculty of Applied Science, Universiti Teknologi MARA Sabah Branch, Kota Kinabalu 88400, Sabah, Malaysia; rosfayanti@gmail.com

[*] Correspondence: fpchee06@ums.edu.my (F.P.C.); ghoshbab@ums.edu.my (B.K.G.)

**Highlights:**

1. The research is focused on analyzing the structural properties of SiGe alloys with different compositions, with a particular focus on the effect of annealing temperature on the quality of the crystalline structure.

2. The findings suggest that the quality of the crystalline structure in SiGe alloys improves as annealing temperature increases, and that this is particularly noticeable in alloys with higher percentages of Ge. This indicates that the thermal treatment of SiGe alloys is an important factor in optimizing their properties for various applications.

3. Understanding the structural properties of SiGe alloys is important for a wide range of technological applications, including electronics, optoelectronics, and thermoelectric devices.

**Abstract:** This study investigates the structural and optical characteristics of Silicon Germanium (SiGe) thin films with varying compositions and annealing temperatures for potential use in electronic and optoelectronic devices. $Si_{0.8}Ge_{0.2}$ and $Si_{0.9}Ge_{0.1}$ films were deposited onto a high-temperature quartz substrate and annealed at 600 °C, 700 °C, and 800 °C before being evaluated using an X-Ray Diffractometer (XRD), Atomic Force Microscopy (AFM), and a UV-Vis Spectrometer for structural and optical properties. The results show that increasing the annealing temperature results in an increase in crystalline size for both compositions. The transmittance for $Si_{0.8}Ge_{0.2}$ decreases slightly with increasing temperature, while $Si_{0.9}Ge_{0.1}$ remains constant. The optical band gap for $Si_{0.9}Ge_{0.1}$ thin film is 5.43 eV at 800 °C, while $Si_{0.8}Ge_{0.2}$ thin film is 5.6 eV at the same annealing temperature. XRD data and surface analysis reveal significant differences between the band edges of SiGe nano-structure materials and bulk crystals. However, the possibility of a SiGe nano-crystal large band gap requires further investigation based on our study and related research works.

**Keywords:** SiGe; thin film; nano-crystal; XRD; opto-electronic; alloy

## 1. Introduction

The widespread consumption of coal and petroleum has a devastating impact on global warming and climate change. In order to mitigate these environmental issues, it is essential to generate eco-friendly energy [1,2]. SiGe semiconductor is a transparent material that offers numerous technical and technological advantages for the future. It has a wide range of applications, including uses for solar cells, transistors, photodetectors, and thermoelectric devices. Optical transparency is also required as it allows for light emission through the wavelength. However, the production of SiGe semiconductor requires a unique

production procedure, requiring that it be put on a high-temperature substrate [3,4]. The minimization of optoelectrical losses in solar photovoltaic cells that use several contact approaches is introduced. However, a low resistance transparent barrier with superior electrical performance has yet to be achieved. Therefore, there is significant potential for the development of a low optoelectrical p-contact.

Silicon-doped germanium offers its own stability and efficient spectral utility; this allows for modulation in the optical window through bandgap engineering [5]. Achieving these specialties might be possible by changing the Ge mole fraction in the SiGe material [5]. According to Fathipour et al., the addition of Ge mole fraction in the c-Si thin film could reduce the bandgap and increase the absorption coefficient of the thin film. On the other hand, the addition of SiGe mole fraction in the composition might affect the structural and optical characterization of the thin film [6]. Increasing the Ge content in a composition may be beneficial due to its stability and improved performance [7]. Additionally, SiGe alloy MOSFET stressors are utilized to induce strain into the Si and Ge channels [8]. The production of nanocomposites is a viable method for lowering SiGe alloys' lattice thermal conductivity [9].

It has been reported that lower deposition temperatures are more effective in improving carrier lifetimes when compared to higher temperatures due to less substantial degradation of Si characteristics [10]. CdTe solar cells have been found to exhibit reduced efficiency with increasing temperature, while mcSi and Si modules have been shown to have higher efficiency when compared to CdTe solar cells [4–11]. Moreover, SiGe thin film is well-known for its higher transparency and exhibits superior photo-electrical properties for future transparent solar cells. In addition, these similar complex material systems have been successfully addressed by ab initio calculations, including the structural and growth aspects [12].

The deposition of SiGe thin film is achieved using the Radio Frequency sputtering method. This method is considered to be the most effective method for depositing a wide range of thin film materials under vacuum conditions [13]. Previous research has found that $Si_{0.5}Ge_{0.5}$ grown by chemical vapor deposition exhibits an optical bandgap of approximately 1 eV, with high defect concentration at lower growth temperatures [14]. Si solar cell is realized to have the best module efficiency, with both chemically and electrically appealing designs for the aSi:H buffer and emitter, as well as metal oxide back passivation [15]. In this research, the structural and optical characterization for both $Si_{0.8}Ge_{0.2}$ and $Si_{0.9}Ge_{0.1}$ were analyzed.

## 2. Experimental Details

To prepare the quartz glass substrate for the sputtering process, it was first cleaned in an ultrasonic bath for 15 min using a combination of distilled water, ethanol, and acetone. The quartz substrate was then rinsed with distilled water and blown dry with an inert gas (nitrogen gas). $Si_{0.8}Ge_{0.2}$ and $Si_{0.9}Ge_{0.1}$ layers were sputtered on the cleaned quartz substrate using a Radio Frequency sputtering machine for about 30 min. Prior to sputtering, the quartz substrate was heated to 900 °C in a furnace. The sputtering process was carried out under the following conditions: a sputtering power of 98 W, a working pressure of 3 mTorr, and a temperature of 900°. A SiGe target was positioned parallel to the substrate to ensure even distribution. The process involved introducing argon gas at a low pressure of less than 0.4 mTorr, causing the gas atoms to become ionized and form plasma with the aid of electrons and neutral atoms. The plasma was then deposited onto the quartz substrate, resulting in the formation of a thin film. Table 1 shows the thin film composition and target parameters used in the deposition process.

**Table 1.** Deposition composition and parameters.

| Thin Film Composition | • Silicon Germanium (Si 90%, Ge 10%) wt% <br> • Silicon Germanium (Si 80%, Ge 20%) wt% |
|---|---|
| Material purity (Sputter target) | 99.99% |
| Target size | 3" diameter × 0.125" thickness |
| Substrate | Quartz Glass |
| Annealing temperature | 600 °C, 700 °C and 800 °C |
| Sputtering power | 98 W |

The sputtering time was kept constant throughout the process to ensure a constant growth thickness of 300 nm for both SiGe compositions. Following the sputtering process, all the samples were annealed inside a furnace for 30 min at varying temperatures of 600 °C, 700 °C, and 800 °C for both SiGe compositions. A Bruker Multimode 8-HR, an atomic force microscope, was used to analyze and study the surface and morphological properties of the deposited SiGe thin film. Additionally, the optical properties of the samples were observed using a Lambda EZ210 UV-Vis spectrometer to measure the optical transmittance and calculate the optical bandgap of the samples.

### 3. Result and Discussion

#### 3.1. Structural Properties of $Si_{0.8}Ge_{0.2}$ and $Si_{0.9}Ge_{0.1}$ Thin Film

XRD data has been analyzed and there are two peaks observed at 64.4° (4 0 0) and 77.5° (3 3 1) for both $Si_{0.8}Ge_{0.2}$ and $Si_{0.9}Ge_{0.1}$ as in Figure 1 [16,17]. From the XRD result in Figure 1, it can be seen that there is some shifting happening at 64.4°; this might be due to lattice mismatch and the sample stage.

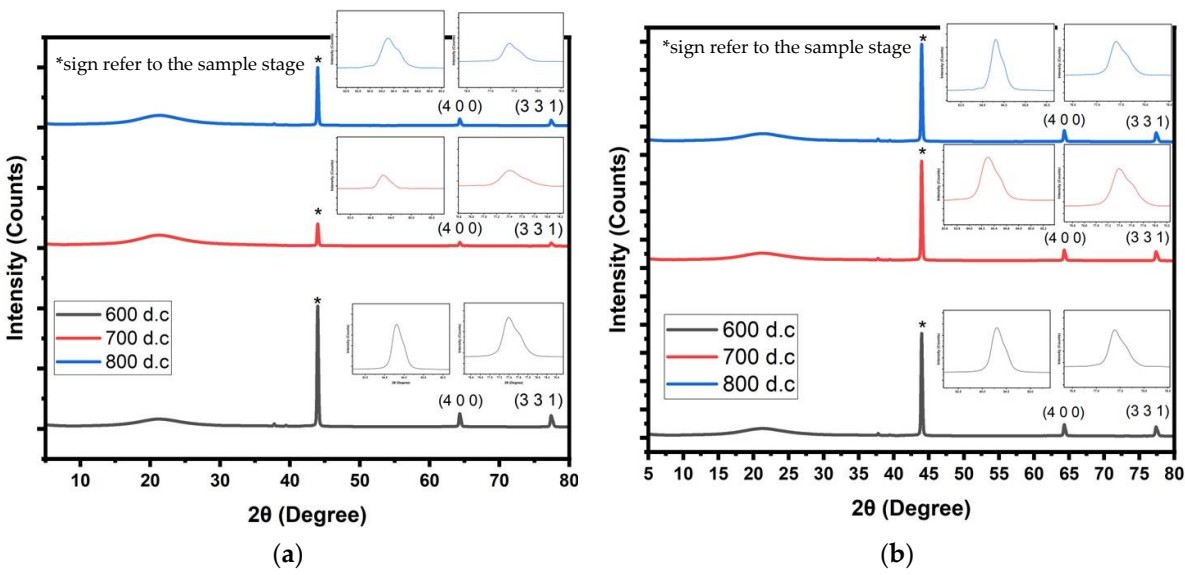

**Figure 1.** XRD result for both (**a**) $Si_{0.8}Ge_{0.2}$ and (**b**) $Si_{0.9}Ge_{0.1}$ composition.

The interplanar distance of the film samples or the lattice parameter was calculated using Equation (1).

$$d_{hkl} = \frac{h^2}{a^2} + \frac{k^2}{b^2} + \frac{l^2}{c^2} \qquad (1)$$

where *h*, *k*, and *l* are the Miller indices. From the XRD data, the increase in annealing temperature causes lattice distortion. It enhances crystal defect, as proven by the micro-strain result in Table 2 for $Si_{0.8}Ge_{0.2}$ composition for 700 °C and 800 °C annealing temperature [18].

The crystallite size for both compositions in $Si_{0.8}Ge_{0.2}$ and $Si_{0.9}Ge_{0.1}$ are calculated using the Scherrer Equation, as stated in Equation (2).

$$D = \frac{k\lambda}{\beta \cos \vartheta} \qquad (2)$$

where $D$ is crystalline size, $k$ is the Scherrer constant, $\lambda$ is the wavelength of x-ray source, $\beta$ is the FWHM in radian, and $\theta$ is the peak position in XRD analysis. The crystallite size is observed to be in nanoscale; this follows the XRD peak in Figure 1 [19]. The micro-strain of the crystallite has been calculated using the micro-strain equation as stated in Equation (3).

$$\varepsilon = \frac{\beta}{4 \tan \theta} \qquad (3)$$

where $\varepsilon$ is the micro-strain in radian, $\beta$ is the line broadening at FWHM in radian, and $\theta$ is the Bragg's angle in degree, which is half of the $2\theta$ value in the XRD data.

**Table 2.** The structural properties for XRD data.

| Composition | | $Si_{0.8}Ge_{0.2}$ | | | | $Si_{0.9}Ge_{0.1}$ | | |
|---|---|---|---|---|---|---|---|---|
| Parameter | Crystallite Size (nm) | Micro-Strain, $\times 10^{-3}$ (Rad) | Lattice Constant | Peak Position | Crystallite Size (nm) | Micro-Strain, $\times 10^{-3}$ (radian) | Lattice Constant | Peak Position |
| 600 °C | 35.79 | 1.92 | 5.58002 | 64.4°, 77.5° | 35.63 | 1.91 | 5.57932 | 64.4°, 77.5° |
| 700 °C | 31.41 | 143.76 | 5.58086 | 64.4°, 77.5° | 22.98 | 1.92 | 5.58102 | 64.4°, 77.5° |
| 800 °C | 36.78 | 165.49 | 5.58057 | 64.4°, 77.5° | 36.28 | 0.97 | 5.58063 | 64.4°, 77.5° |

Table 2 reveals a discrepancy between the crystallite size from the XRD data and the grain size from the AFM data. Specifically, the crystallite size increases from 35.79 nm and 36.78 nm at 600 °C and 800 °C for $Si_{0.8}Ge_{0.2}$ composition, respectively, while the grain size obtained from the AFM data increases from 61.08 nm to 74.87 nm over the same temperature range. This can be explained by the fact that the crystallite size is composed of several particles that are coherent diffraction domains in X-ray diffraction and is dependent on the size of the defect-free volume, while grains are volumes inside crystalline materials with a specific orientation and are visualized without taking the degree of structural imperfection into account. The XRD data indicate that an increase in annealing temperature leads to lattice distortion and enhanced crystal defect, as evidenced by micro-strain results in Table 2 for $Si_{0.8}Ge_{0.2}$ composition at 700 °C and 800 °C annealing temperatures [18]. However, strain characteristics are also impacted by changes in composition and lead to an increase in carrier mobility [20]. Surface roughness, particle analysis, and particle density for both $Si_{0.8}Ge_{0.2}$ and $Si_{0.9}Ge_{0.1}$ were also analyzed using Atomic Force Microscopy (AFM). Figure 1 displays the crystalline image of $Si_{0.8}Ge_{0.2}$ and $Si_{0.9}Ge_{0.1}$ at different temperatures. For the $Si_{0.8}Ge_{0.2}$ composition, the lower annealing temperature resulted in a larger grain size than the higher annealing temperature. The AFM analysis in Figure 1 shows that richer grains can be observed at 700 °C and 800 °C when compared to the lower temperature. However, $Si_{0.9}Ge_{0.1}$ composition shows a constant trend in grain size from the AFM analysis, as shown in Figure 2d–f. Both compositions have larger grain sizes at an annealing temperature of 800 °C.

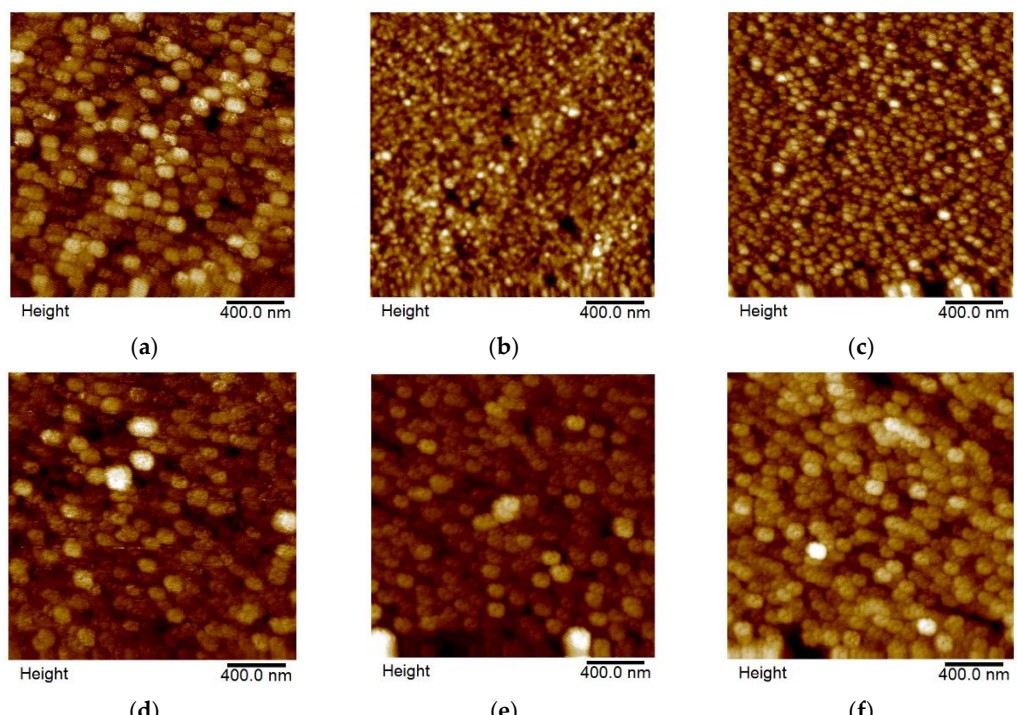

**Figure 2.** AFM analysis for both crystalline $Si_{0.8}Ge_{0.2}$ at (**a**) 600 °C, (**b**) 700 °C, and (**c**) 800 °C, and $Si_{0.9}Ge_{0.1}$ (**d**) 600 °C, (**e**) 700 °C, and (**f**) 800 °C.

The AFM analysis was supported by the XRD data (shown in Figure 3); these data reveal the crystallite size for both SiGe compositions. Based on Figure 3, the highest roughness was observed at 700 °C for each composition, reaching 34.7 nm, and then decreased to 27.7 nm as the temperature increased. The trend was consistent for both compositions. This tendency justifies the dislocation rearrangement and removal of point defect during recovery, which could considerably lower the amount of stored energy [21,22] and weakens the bonds of the material [23,24]. Samples annealed at high temperatures have more energy trapped inside the matrix due to insufficient time for recovery between grains [21]. This may also be caused by insufficient recovery and faster grain growth rates due to higher stored energy within the samples. Table 3 shows the summary of structural properties, including particle density, for both $Si_{0.8}Ge_{0.2}$ and $Si_{0.9}Ge_{0.1}$ compositions.

**Table 3.** The summary of structural properties for both SiGe compositions using AFM analysis.

| Composition | $Si_{0.8}Ge_{0.2}$ | | | $Si_{0.9}Ge_{0.1}$ | | |
|---|---|---|---|---|---|---|
| Parameter | Grain Size (nm) | RMS Roughness (nm) | Particle Density ($\mu m^{-2}$) | Grain Size (nm) | RMS Roughness (nm) | Particle Density ($\mu m^{-2}$) |
| 600 °C | 61.078 | 20.3 | 19.961 | 69.359 | 23.9 | 24.897 |
| 700 °C | 70.895 | 28.6 | 51.401 | 87.978 | 34.7 | 17.889 |
| 800 °C | 74.868 | 26.4 | 52.486 | 97.689 | 27.7 | 19.868 |

Based on Table 3, the particle density for $Si_{0.8}Ge_{0.2}$ increases with the annealing temperature, whereas for $Si_{0.9}Ge_{0.1}$, the particle density decreases with increasing temperature. Particle density was determined using an Atomic Force Microscopy (AFM) nano-scope analyzer from Bruker [25]. Among the compositions, $Si_{0.8}Ge_{0.2}$ has the highest particle density, with a value of 52.486 $\mu m^{-2}$ at 800 °C, which is higher than that of $Si_{0.9}Ge_{0.1}$. This observation suggests that $Si_{0.8}Ge_{0.2}$ has a more stable structure than $Si_{0.9}Ge_{0.1}$, consistent with the reported decrease in lattice parameter as Ge concentration increases [26]. These results support the benefits of higher particle density, such as in the double stack structure,

which may increase cation disordering due to high thermal energy in the nanocrystalline system [27]. For both compositions, the highest particle roughness was observed at 700 °C, with values of 28.6 nm and 34.7 nm for $Si_{0.8}Ge_{0.2}$ and $Si_{0.9}Ge_{0.1}$, respectively.

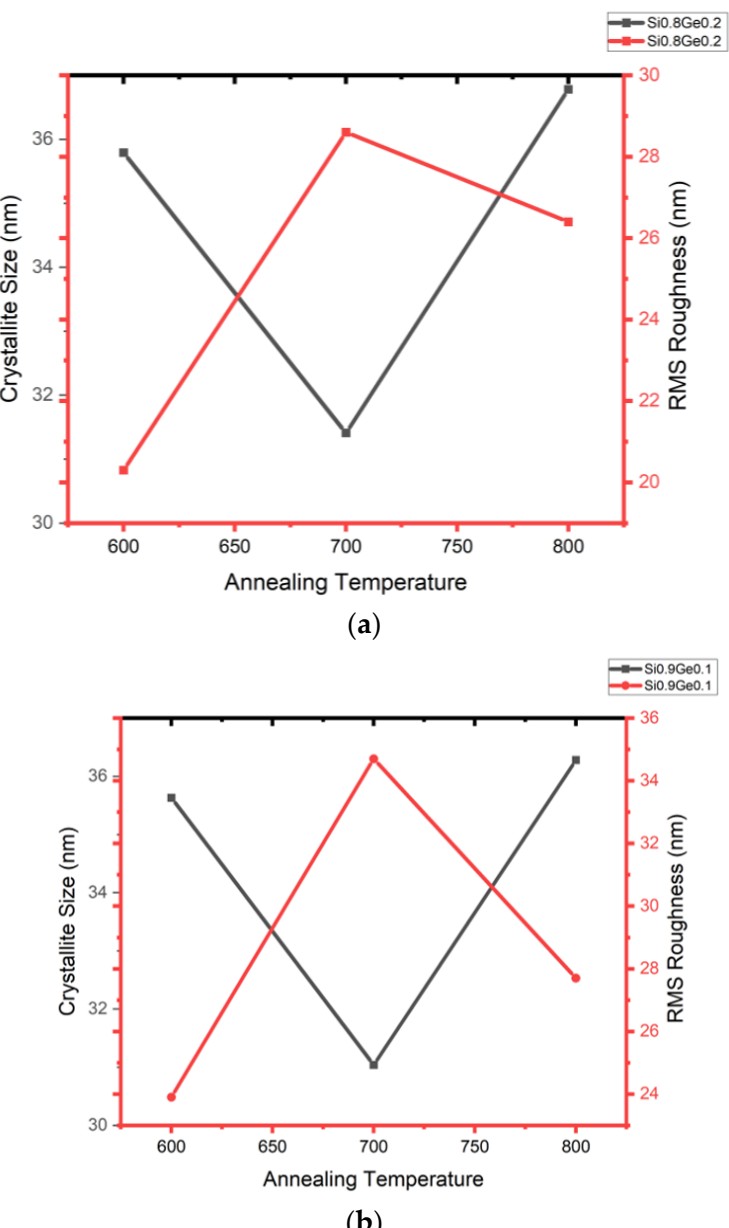

**Figure 3.** The crystallite size and RMS roughness graph against annealing temperature for both (**a**) $Si_{0.8}Ge_{0.2}$ and (**b**) $Si_{0.9}Ge_{0.1}$.

### 3.2. Optical Properties Analysis for Both $Si_{0.8}Ge_{0.2}$ and $Si_{0.9}Ge_{0.1}$ at Different Annealing Temperature

The transmittance and absorbance graph from our UV-Vis analysis has been plotted for both materials at different annealing temperatures, as shown in Figure 4. At 600 °C, the transmittance of $Si_{0.8}Ge_{0.2}$ at 850 nm wavelength is 87.87%, while the transmittance for $Si_{0.9}Ge_{0.1}$ remains constant at 87.90% across all annealing temperatures. The observed trend can be attributed to the roughness of grain boundaries in $Si_{0.9}Ge_{0.1}$, which might be caused by the rapid annealing process employed for this composition [28,29].

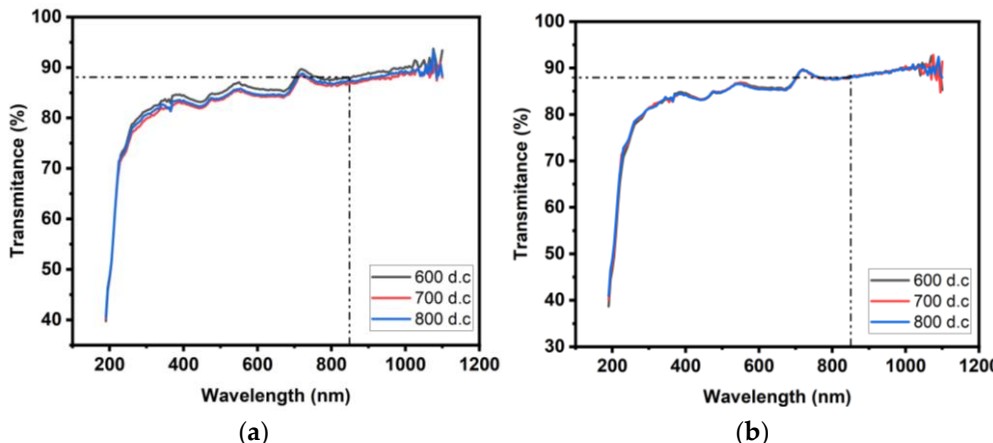

**Figure 4.** Transmittance graph for (**a**) $Si_{0.8}Ge_{0.2}$ and (**b**) $Si_{0.9}Ge_{0.1}$ at 600 °C, 700 °C and 800 °C.

The absorbance values obtained can be utilized to estimate the band gap energy values of the formed SiGe films. According to the absorbance data, the optical bandgap for both compositions was calculated using Equation (4) [30].

$$\alpha h v \approx \left( h v - E_g \right)^{1/2} \tag{4}$$

where $\alpha$ is the absorption coefficient, $hv$ is the incident photon energy, and $E_g$ refer to the optical bandgap [31]. Table 4 shows the summary of optical bandgap and transmittance for both compositions of SiGe.

**Table 4.** The summary of transmittance at 850 nm wavelength and bandgap for both SiGe compositions from the absorption graph.

| Composition | $Si_{0.8}Ge_{0.2}$ | | $Si_{0.9}Ge_{0.1}$ | |
|---|---|---|---|---|
| Optical Parameter | Transmittance (%) | Bandgap (eV) | Transmittance (%) | Bandgap (eV) |
| 600 °C | 87.87 | 5.505 | 87.9 | 5.528 |
| 700 °C | 86.86 | 5.380 | 87.9 | 5.558 |
| 800 °C | 87.28 | 5.595 | 87.9 | 5.429 |

The optical bandgap was found to be approximately 5.505 eV for $Si_{0.8}Ge_{0.2}$ at 600 °C, while the bandgap for $Si_{0.9}Ge_{0.1}$ is slightly higher (5.528 eV) at the same temperature. At 800 °C, $Si_{0.8}Ge_{0.2}$ composition has a higher bandgap (5.595 eV), while $Si_{0.9}Ge_{0.1}$ has the lowest bandgap (5.429 eV). From the optical bandgap summary in Table 4, an annealing temperature of 800 °C was found to be the most effective for $Si_{0.9}Ge_{0.1}$, resulting in a lower bandgap and, therefore, a lower recombination rate [32]. As for $Si_{0.8}Ge_{0.2}$ thin film, the structural modifications for increased quantum confinement can be attributed to the effects where the optical bandgap increases as the annealing temperature rises [33–35]. Overall, it was observed that the optical bandgap increases as the grain size of the nanocrystalline state increases, as seen in XRD data in Figure 1 and Table 3 [36]. It is realized that for nanocrystalline SiGe thin film, the discrepancy from the bandgap approximation might be explained by the bands' nonparabolicity as a result of band-folding and state mixing, while holes and electrons cannot be transferred to tiny clusters [37,38]. Furthermore, explorations between the bandgap of bulk and nanocrystalline SiGe thin film have reported that the dependence on Ge concentration is consistent with the results presented in Table 3 [36]. The invariance of the shape can be described by Equation (5) [36].

$$E_g(d, x) = E_0^{NC} + 4.58/d^{1.25} \tag{5}$$

Figure 4 shows the transmittance data for both SiGe compositions. The absorption coefficient and bandgap for $Si_{0.8}Ge_{0.2}$ show a decreasing trend at 700 °C, followed by a rapid increase at 800 °C. This indicates that $Si_{0.8}Ge_{0.2}$ has a greater ability to control how far light of a specific wavelength can pass into a substance before being absorbed. On the other hand, $Si_{0.9}Ge_{0.1}$ exhibits a decreasing trend in bandgap across the temperature range, but its absorption coefficient increases. Figure 5 plots the (ahv)2 versus energy graph for both compositions to further investigate the relationship between absorption coefficient and energy. From Peng et al., the absorption edge for pure quartz glass shows a lower absorption edge than in this study. The absorption peak observed in Figure 6 indicates that the SiGe thin films are deposited on top of the quartz glass [39]. Overall, these results suggest that the optical properties of SiGe thin films are highly dependent on the composition and annealing conditions, which could have important implications for their potential applications in optoelectronics and other fields.

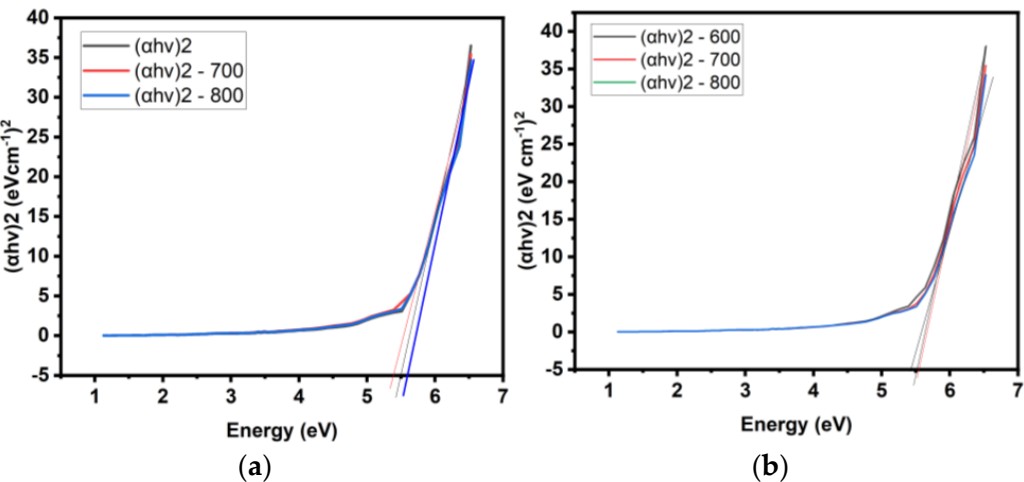

**Figure 5.** The energy vs. $(ahv)^2$, band gap graph for (**a**) $Si_{0.8}Ge_{0.2}$ and (**b**) $Si_{0.9}Ge_{0.1}$ composition.

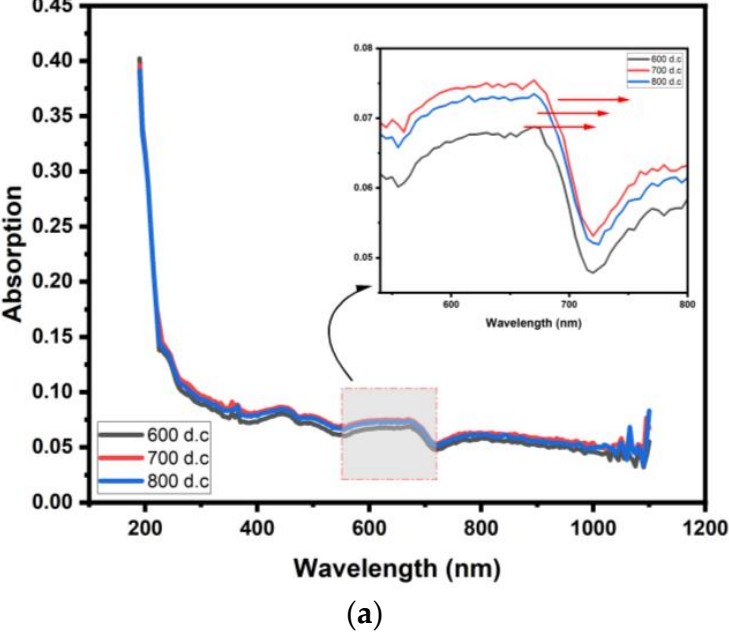

(**a**)

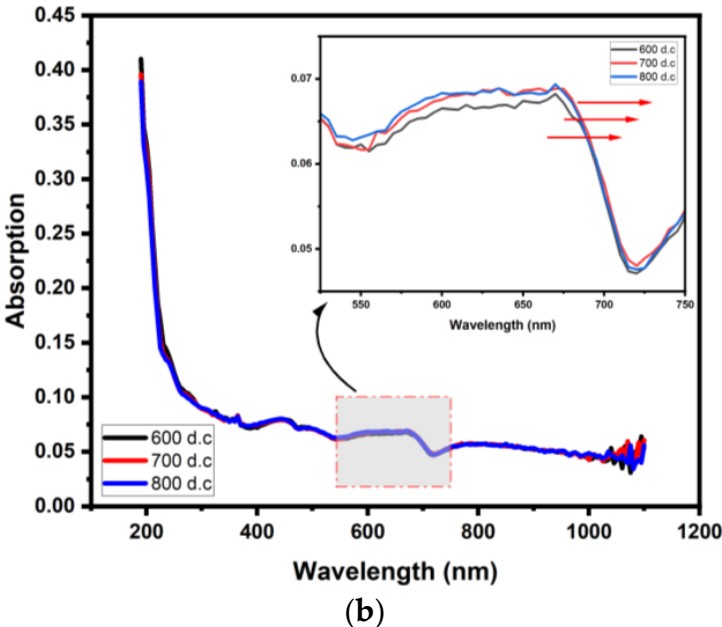

**(b)**

**Figure 6.** Absorbance shifts for (**a**) $Si_{0.8}Ge_{0.2}$ and (**b**) $Si_{0.9}Ge_{0.1}$ thin film.

Photoluminescence is the light emission from any matter after photon absorption, also known as UV-Vis spectroscopy. This process involves the absorption of photons by a material, followed by re-emission of photons at lower energies with a smaller number of photons. Photoluminescence can be analyzed with the absorbance graph from UV-Vis data [40,41]. Figure 6 shows the absorbance peaks for both $Si_{0.8}Ge_{0.2}$ and $Si_{0.9}Ge_{0.1}$ shift towards the right side; this is observed as a red shift. These observations are consistent with the AFM data for $Si_{0.8}Ge_{0.2}$ and $Si_{0.9}Ge_{0.1}$ at 800 °C (74.5 nm and 97.7 nm, respectively). At high doping concentrations, the band gap for both $Si_{0.8}Ge_{0.2}$ and $Si_{0.9}Ge_{0.1}$ thin films is reduced due to the convergence of the donor and conduction bands; these are the effects of red shift in the optical absorption [42–45].

## 4. Conclusions

This study investigated the impact of annealing temperature on the structural and optical properties of$Si_{0.8}Ge_{0.2}$ and $Si_{0.9}Ge_{0.1}$ compositions fabricated using RF sputtering. Both compositions exhibited significant changes in their optical properties, with $Si_{0.8}Ge_{0.2}$ showing greater variation than $Si_{0.9}Ge_{0.1}$. The transparency analysis revealed that the transparency of $Si_{0.8}Ge_{0.2}$ decreased with increasing temperature, while the transparency of $Si_{0.9}Ge_{0.1}$ remained constant at around 87.9%. The difference in transparency between the two compositions was found to be 0.1%. The structural stability of $Si_{0.8}Ge_{0.2}$ was attributed to its higher grain size and particle density when compared to $Si_{0.9}Ge_{0.1}$, which had lower values for these properties. Overall, $Si_{0.9}Ge_{0.1}$ was found to be the optimal composition in this study due to its higher transparency, larger grain size, and crystallite size at 800 °C. XRD and surface analysis revealed that SiGe nanostructure materials exhibited a larger bandgap than bulk crystals. These findings suggest that SiGe with a larger bandgap may be advantageous in ultraviolet photodetector applications.

**Author Contributions:** Conceptualization, B.K.G., F.P.C. and S.N.; methodology, S.N.; software, M.I.R.; validation, B.K.G., F.P.C. and M.D.; formal analysis, S.N.; investigation, S.N; resources, S.N.; data curation, R.R., F.J., M.I.R., M.D. and S.N.; writing—original draft preparation, S.N.; writing—review and editing, S.N.; visualization, R.R.; supervision, B.K.G. and F.P.C.; project administration, B.K.G.; funding acquisition, F.P.C. and B.K.G. All authors have read and agreed to the published version of the manuscript.

**Funding:** This research was funded by Ministry of Higher Education Fundamental Research Grant Scheme (FRGS/1/2020/STG07/UMS/02/1) and Universiti Malaysia Sabah (SDK0311-2020). The APC was funded by Universiti Malaysia Sabah.

**Data Availability Statement:** Not applicable.

**Conflicts of Interest:** The authors declare no conflict of interest.

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
