# Peer review of "Composition Dependence Structural and Optical Properties of Silicon Germanium (SiχGe1−χ) Thin Films"

_crystals, doi:10.3390/cryst13050791_

Round 1

Reviewer 1 Report

The paper reported the deposition of SiGe alloys by RF sputtering method with different annealing temperature and characterized the as-synthesized films by XRD, AFM and UV-vis spectrometer. I don’t think the paper can be published in its present form. Major revision is needed.

1. The band gaps determined from the absorption spectra were totally wrong. The authors used the absorption edge from quartz glass substrate to determine the band gap. In fact, the SiGe alloys cannot have such a wide band gap.

2. The thickness of the films should be verified by AFM measurements.

3. The insets of figure 1 can not be clearly seen.

4. Why Si0.8Ge0.2 and Si0.9Ge0.1 have the same lattice constant?

5. Why does the higher particle density suggests a more stable structure?

Author Response

All the corrections were made accordingly. 

Reviewer 2 Report

This manuscript focuses on the “Composition Dependence Structural and Optical Properties of Silicon Germanium (SicGe1-c) Thin Films” for applications in electronics, optoelectronics, and thermoelectric devices. The topic is of great interest and authors pointed out some differences and improvement in optical properties by utilizing the annealing treatment at 600, 700, and 800 °C. However, the manuscript does not provide any new information, all these details have been well-known in the literature. Also, I don’t understand why they selected only Si0.9Ge0.1 and Se0.8Ge0.2 compositions. The overall results are normal, no new findings, considering the reputation of the crystal journal, I would not recommend this manuscript for publication.

Author Response

(The authors gave the same response as above.)

Reviewer 3 Report

This manuscript reports on the structural and optical properties of silicon&germanium thin films. The authors put the emphasis on the dependence of these properties from the composition of the films as well as on the deposition conditions, most particularly, the temperature. Material system is interesting for variety of applications such as optoelectronic and thermo-electric devices. It is also not very well studied but attracts growing research interest. Appropriate characterization methods especially XRD, AFM and UV-Vis Spectroscopy are employed thus providing detailed knowledge to the obtained structures. Understanding the phenomena associated with structural and structural dependence on composition of the investigated material concept is the important novelty of the present work. The authors convincingly make conclusions on SiGe material system and how to achieve controlled growth and pre-desired properties.

Thus, besides being very timely, the present work is also much credible and highly original.

The presentation of results is clear and appealing, an easy and consistent read and thus can be very helpful for the interested community to make choices and benchmarking of the SiGe thin films. Useful and employable solutions can be extracted from the rich characterization results as well as from the discussion provided, while they are well presented and given an excellent context toward possible applications.

All in all, this work certainly represents a very valuable contribution with possible wider impact to the field.

The authors chose an adequate structure of the manuscript. Also, concise, and nicely illustrated figures and their corresponding analysis are provided.

This work once published would be instructive and suggestive in terms of further studies and with excellent chances be cited by other teams.

There are some minor issues with this already excellent manuscript that will need to be addressed before the manuscript becoming suitable for publication, i.e., it can be considered for publication after a minor revision:

1: Title is grammatically incorrect: “Composition dependence of structural …”. It is also grammatically incorrect in English to begin each word of a title with a capital letter. There should not a full stop at the end of a title. Well written title is important and it may attract even more interest to this excellent work.

2:There are some expressions that are not well chosen, e.g., “ … distracts the bonds of material”  (page 6), the authors want to say “weakens the bonds of the material”.

3: The annealing temperatures should be commented in the context of the thermal stability of the SiGe phases/nanocrystals.

4: In the introduction, the authors miss that material systems of similar complexity have been successfully addressed by ab initio calculations including structural and growth aspects [ACS Nanosci. Au 2023, 3, 1, 84–93, Dalton Transactions 44 (2015) 3356-3366]. Such theoretical approaches should be acknowledged thus providing wider context of the present work.

5: Spell-check and stylistic revision of the paper are necessary. Some long sentences, as well as misspellings, etc., are noticeable throughout the text.

Spell-check and stylistic revision of the paper are necessary. Some long sentences, as well as misspellings, expressions that need rewriting, are noticeable throughout the text.Title should be rewritten (see review report)

Author Response

(The authors gave the same response as above.)

Reviewer 4 Report

Recommendation: Major revision

In this manuscript, the authors reported the structural and optical characteristics of SiGe with varying compositions and annealing temperatures. However, in my opinion, the present manuscript needs some improvement in order to be published in the journal of Crystals.

1. As a research article, the present experimental results are limited.

2. Why did the authors select only Si0.8Ge0.2 and Si0.9Ge0.1?

3. There are various grammatical mistakes, which should be corrected.

4. The numbering of references throughout the manuscript should be corrected. In the introduction section the first reference began with  [27] .

5. Please check carefully the manuscript and put the related reference for all equations.

6. Related references for all statements should be added.

7. In the Introduction section the authors wrote: " Reported that lower deposition temperatures are more effective in improving carrierlifetimes compared to higher temperatures due to less degradation of Si characteristics. " Some related publications should be referred.

8. In all Figures "a)" and "b)" "c)" and "d)" should be up or downside the figure with the same format.

 There are various grammatical mistakes, which should be corrected. Moreover, the numbering of references throughout the manuscript should be corrected. In the introduction section the first reference began with  [27] .

Author Response

(The authors gave the same response as above.)

Round 2

Reviewer 1 Report

Still about the band gap. If the authors insist the materials have large bandgaps, how to exclude the contribution from the quartz substrate because quartz glass has a similar absorption edge?

Author Response

Thank you for the comment. The manuscript is improved accordingly. 

Reviewer 2 Report

The authors have addressed some of my concerns in the response letter. It appears that this study is still useful and can be published in Crystals Journal.

Author Response

Thank you for the comment. Your suggestions are highly appreciated.